# Artificial Intelligence Based Algorithms for Prostate Cancer Classification and Detection on Magnetic Resonance Imaging: A Narrative Review

**DOI:** 10.3390/diagnostics11060959

**Published:** 2021-05-26

**Authors:** Jasper J. Twilt, Kicky G. van Leeuwen, Henkjan J. Huisman, Jurgen J. Fütterer, Maarten de Rooij

**Affiliations:** Department of Medical Imaging, Radboud University Medical Center, P.O. Box 9101, 6500 HB Nijmegen, The Netherlands; kicky.vanleeuwen@radboudumc.nl (K.G.v.L.); henkjan.huisman@radboudumc.nl (H.J.H.); jurgen.futterer@radboudumc.nl (J.J.F.); maarten.derooij@radboudumc.nl (M.d.R.)

**Keywords:** artificial intelligence, machine learning, radiomics, deep learning, prostate neoplasms, computer-aided diagnosis, magnetic resonance imaging

## Abstract

Due to the upfront role of magnetic resonance imaging (MRI) for prostate cancer (PCa) diagnosis, a multitude of artificial intelligence (AI) applications have been suggested to aid in the diagnosis and detection of PCa. In this review, we provide an overview of the current field, including studies between 2018 and February 2021, describing AI algorithms for (1) lesion classification and (2) lesion detection for PCa. Our evaluation of 59 included studies showed that most research has been conducted for the task of PCa lesion classification (66%) followed by PCa lesion detection (34%). Studies showed large heterogeneity in cohort sizes, ranging between 18 to 499 patients (median = 162) combined with different approaches for performance validation. Furthermore, 85% of the studies reported on the stand-alone diagnostic accuracy, whereas 15% demonstrated the impact of AI on diagnostic thinking efficacy, indicating limited proof for the clinical utility of PCa AI applications. In order to introduce AI within the clinical workflow of PCa assessment, robustness and generalizability of AI applications need to be further validated utilizing external validation and clinical workflow experiments.

## 1. Introduction

With a worldwide estimation of 1.4 million new cases in 2020, prostate cancer (PCa) is the second most common malignancy among men worldwide [1]. Despite the high prevalence of PCa, PCa related deaths account for merely 10% of all cancer deaths with five-year survival rate exceeding 98% for all PCa stages combined [2]. Considering the high PCa prevalence and low mortality rate, accurate differentiation between aggressive and non-aggressive PCa is of high importance to decrease overdiagnosis and overtreatment. Artificial intelligence (AI) techniques may have the potential to highlight important characteristics indicative of disease and therefore could provide significant aid in PCa management [3].

In 2018 and 2019, several large prospective trials concluded that the use of magnetic resonance imaging (MRI) prior to biopsy increases the detection of (more aggressive) clinically significant (cs)PCa, while decreasing detection of (non-aggressive) clinically insignificant (cis)PCa compared to transrectal ultrasound guided biopsy [4,5,6,7]. For this reason, multiparametric (mp)MRI has been included in the guidelines of the European Association of Urology (EAU) to be performed prior to biopsy [8]. It is recommended to use the Prostate Imaging and Reporting and Data System (PI-RADSv2.1) to report prostate MRI [9]. Suspicious lesions are graded from highly unlikely to highly likely for csPCa using a five-point Likert scale.

Due to the upfront role of mpMRI in the diagnostic pathway of PCa, the workload of prostate MRI examinations increases. Reporting these exams, however, requires substantial expertise and is limited by a steep learning curve and inter-reader variability [10,11,12]. Computer-aided detection and/or diagnosis (CAD) applications using AI may have a role in overcoming these challenges and aid in improving the workflow of prostate MRI assessment. Before AI-CAD applications for prostate MRI can be introduced within a clinical workflow, current applications described within literature and corresponding evidence for its potential use need to be investigated.

In this review we provide an overview of studies describing AI algorithms for prostate MRI analysis from January 2018 to February 2021, in which we differentiate applications for lesion classification and lesion detection for PCa. The study methodologies, data characteristics, and level of evidence are described. We furthermore review commercially available CAD software for PCa and discuss its clinical applications.

## 2. Background

### Machine Learning and Deep Learning Approaches

AI encompasses various subsets of learning techniques and algorithms. Machine learning (ML) is a subset within AI comprising algorithms that learn and predict specific tasks without explicit programming. For a long time, these ML techniques have served as the main pipeline for CAD applications [13]. Contrary to classical rule-based algorithms, ML is capable of learning and improving its task over time while being exposed to large and new data [14].

ML algorithms learn and predict by extracting and utilizing features [15]. In the field of prostate MRI, these features are mainly extracted from T2-weighted sequences and DWI with ADC maps and may additionally be combined with clinical parameters such as serum prostate-specific antigen (PSA) level and PSA density (PSAd). An expanding field used for image feature selection is that of radiomics. Radiomics concern the extraction of quantitative features from a region of interest (ROI), such as an annotation of a suspicious lesion, to describe the distinctive attributes of the ROI. Both semantic features, such as size and shape, as agnostic features, such as textures, are mined. The most significant features are selected and used in the learning task of the ML algorithm [16].

In more recent years, a particular subset of ML, deep learning (DL), gained popularity in CAD [13]. In contrast to classical ML algorithms, DL does not require prior feature extraction as the algorithm learns to extract complex and abstract features during training [14]. DL algorithms can be divided into typical algorithms utilizing one-dimensional feature input, or convolutional neural networks (CNN), utilizing two- and three-dimensional feature input, such as prostate mpMRI sequences. CNNs are often utilized within medical image analysis [17]. Although DL algorithms may be implemented without prior feature selection, these algorithms are limited by the need for extensive data for training. In addition, due to their complex architecture, DL algorithms are less transparent and difficult to interpret, which impedes widespread application [18].

## 3. Materials and Methods

The Pubmed and Cochrane libraries were searched for studies describing ML algorithms for the characterization, detection, and grading of PCa on MRI. The search was limited to articles written in English from 2018 to February 2021 using combined terms: artificial intelligence, machine learning, prostate cancer, magnetic resonance imaging, and corresponding synonyms for each term. The search was limited to these years to retrieve articles most representative for the current research field. Additional references were identified by manual search in the reference list of included papers. Duplicates, reviews, conference abstracts, preceding articles of described algorithm, and articles not related to the topic were excluded (Figure 1).

For categorization between various AI-CAD algorithms, studies were categorized within two common tasks [13,19]:*1.* *Lesion classification algorithms*, i.e., *Computer-Aided Diagnosis (CADx)*Within the first group we included algorithms that classify manually annotated regions, such as lesion segmentations. We discriminate between two-class classification algorithms, utilizing either ML or DL, and multi-class classification algorithms.*2.* *Lesion detection algorithms*, i.e., *Computer-Aided Detection (CADe)*The second group included algorithms that detect and localize PCa lesions and provide the user with probability maps, segmentations, and/or attention boxes as output. We discriminate between algorithms providing two-class detection and multi-class detection.

For all studies, AI algorithm characteristics, MRI sequences used, study design and cohort size, ground truth for PCa, and performance were extracted. Studies were graded using an adaptation of the hierarchical model for diagnostic imaging efficacy from Fryback and Thornbury, applicable for assessment of AI software in clinical practice (Table 1) [20,21].

Secondly, a search for commercially available CAD software for PCa was performed to investigate current available products for clinical application. Applications were included if CAD was suited for prostate MRI assessment and received Food and Drug Administration (FDA) clearance and/or European Conformity (CE) marking. For included applications, key features, market date, and literature evidence were assessed.

## 4. AI Algorithms for Prostate Cancer Classification and Detection

In total, 59 studies were included in this review (Figure 2). Thirty-nine articles (66%) described lesion classification algorithms. Of these, 35 articles (59%) described two-class lesion classification with 25 (42%) articles using an ML and 10 articles (17%) a DL approach. Four articles (7%) were included for multi-class lesion classification. The 20 remaining articles (34%) described lesion detection algorithms, with 17 studies (29%) for two-class lesion detection and 3 studies (5%) for multi-class lesion detection. Additionally, 6 commercially available AI applications for prostate MRI with either FDA clearance and/or CE marking were identified. In the next sections, topics will be summarized according to each category.

### 4.1. Lesion Classification (CADx)

In recent years, numerous ML and DL algorithms have been described for classification (CADx) of suspicious prostate lesions on MRI. Its task is to classify a manually annotated ROI in two or multiple classes, such as malignant versus benign tissue, classification between csPCa and cisPCa, or multi-class classification according to lesion aggressiveness (histopathological grading) or likelihood of csPCa (PI-RADS). Due to the different AI architecture of ML and DL and the large number of included studies, we describe the two-class lesion classification for ML and DL approaches separately.

#### 4.1.1. Two-Class Lesion Classification with Machine Learning

In total, twenty-five studies described two-class lesion classification with a ML approach (Table 2).

Most of these algorithms follow a similar workflow (Figure 3). MR exams are used as input, either multiparametric or single sequence MR. Suspicious regions are manually or semiautomatically annotated by expert readers and used to extract image features. Image features comprise semantic features such as size, shape, and vascularity and agnostic features which describe the heterogeneity of the ROI through quantitative descriptors [16]. As shown in Table 2, image features may be extended with clinical variables such as PSAd. Subsequently, features with a strong relationship with the output labels are selected and used in the ML classification model. The output of the algorithm is a prediction score for two-classes, such as malignant versus benign lesions, for annotated ROIs. Included studies comprised cohort sizes ranging from 20 to 381 patients (median = 129). The gold standard for malignant lesions was obtained via prostate biopsy (19/25 (76%)) or after radical prostatectomy (7/25 (28%)). For most studies, lesion classification was based on either classification between malignant (ISUP ≥ 1) and benign lesions or csPCa (ISUP ≥ 2) vs. cisPCa (ISUP 1).

Only a limited number of studies involved multicenter data (4/25 (16%)), whereas the remaining studies utilized retrospectively collected data from a single center (21/25 (84%)). In nine studies, the performance was assessed with cross-validation methods due to a limited study cohort size. Sixteen studies assessed performance on unseen data. Kan et al., Viswanath et al., and Zhang et al. used data from a different institution for validation, and Dinh et al. and Transin et al. repeated a validation of a prior validated algorithm in a new and external cohort, providing assessment on the generalizability of the algorithm [29,33,41,43,47]. In the work of Kan et al., validation on an internal test set yielded a per lesion AUC for PCa characterization of 0.83. When tested on an external cohort, the per lesion performance decreased to an AUC of 0.67, indicating the importance of external validation for robust performance assessment [33,49].

Most of the included studies for ML based two-class lesion classification solely described the stand-alone performance of the algorithm and did not investigate the influence of CADx in a (prospective) clinical workflow, resulting in a level 2 efficacy (stand-alone performance; see also Table 1). To aid in performance interpretation, ten out of twenty-five studies compared their algorithm with visual scoring by radiologists. For example, Antonelli et al. compared the performance of the algorithm with the assessment of three radiologists to identify Gleason 4 components in suspicious MRI lesions. The algorithm yielded a higher sensitivity at a 50% threshold for lesion classification in the peripheral zone (0.93) compared with the mean sensitivity of the three radiologists (0.72) [24].

In order to investigate the added clinical value of AI based lesion classification, Xu et al. and Zhang et al. introduced decision curve analysis (DCA) using retrospective data. DCA analysis is utilized to assess the clinical utility and additional benefit for a prediction algorithm e.g., assessment of an algorithm to reduce the number of unnecessary biopsies. As a result, a simulated impact on patient management is provided which benefits the interpretation of its clinical utility (efficacy level 4) [50]. Both Xu et al. and Zhang et al. showed that, compared to the treat-all-patients scheme or the treat-none scheme, ML algorithms could improve net benefit if the threshold probability of a patient or doctor was higher than 10% [46,47].

#### 4.1.2. Two-Class Lesion Classification with Deep Learning

In total, ten studies were included in which DL was used for two-class lesion classification (Table 3).

Compared to ML, DL does not require feature selection as features are learned during training (Figure 4). An ROI is annotated on MRI. As depicted in Table 3, ROIs encompass patches or volumes around the lesion or prostate gland and may be extended with clinical features. Selected ROIs are fed into a DL classification algorithm, in which features are extracted and one of two classes is predicted for the corresponding input. Alternatively, DL can be combined with ML in which a DL approach is used for feature extraction and a ML algorithm for classification [58]. Of the included studies, cohort size ranged from 18 to 499 patients (median = 278). Ground truth was provided by biopsy (9/10 (90%)) or radical prostatectomy (1/10 (10%)). Six out of the ten studies aimed to characterize benign tissue from csPCa (ISUP ≥ 2) and four studies aimed to classify benign from malignant lesions (ISUP ≥ 1).

Of all studies, only a single study utilized multiple datasets without the use of external data for performance validation. To overcome the limitation of smaller datasets, Chen et al., Yuan et al., and Zhong et al. utilized transfer learning [52,59,60]. With this approach, pretrained algorithms for a different classification task are applied within a different but related learning task and therefore decrease the large labeled data requirement [14]. Chen et al. utilized a pretrained network on diabetic retinopathy diagnosis, which was trained on a dataset of 128,000 images [52]. Zhong et al. showed that higher AUC and accuracy could be achieved with transfer learning (AUC = 0.726, accuracy = 0.723), compared with the DL model without transfer learning (AUC = 0.687, accuracy = 0.702) [60].

Similarly to the ML algorithms, most described DL algorithms in Table 3 provided the stand-alone performance of the algorithm (efficacy level 2). Deniffel et al. and Zhong et al. compared the performance of the algorithm with visual assessment by radiologists improving the interpretability of the algorithm performance [53,60]. Furthermore, Deniffel et al., Song et al., and Takeuci et al. performed DCA to assess clinical utility of DL to avoid unnecessary biopsy [53,55,56]. The algorithm of Deniffel et al. was additionally calibrated prior to performance assessment to match the observed probability of csPCa within the algorithm, to the true probability of csPCa in the population. Uncalibrated algorithm performance may result in reduced clinical net benefit [53,61]. Deniffel et al. showed that the calibrated performance of a CNN model can reduce the number of biopsies as compared to using PI-RADSv2 alone or combined with PSAd [53].

#### 4.1.3. Multi-Class Lesion Classification

Several recent algorithms have introduced multi-class lesion classification, utilizing both conventional ML as DL algorithms, to assess lesion aggressiveness (*n* = 4 studies, Table 4).

Assessment of the aggressiveness is important for PCa management. The histopathological grade is defined by the International Society of Urological Pathology (ISUP) [68]. Patients with cisPCa (often ISUP 2 or lower) are eligible for active surveillance (AS) whereas men with higher grade lesions (ISUP > 2) are advised to undergo invasive treatment, such as radical prostatectomy or radiotherapy [8,69]. Multi-class lesion classification algorithms utilize ML or DL techniques to grade input ROIs in different groups according to lesion aggressiveness (Figure 5). Of the included studies, cohort size ranged from 72 to 112 patients (median = 99) and all studies used prostate biopsy as ground truth.

Abraham et al. classified lesion patches into the five different ISUP categories using a CNN. The algorithm utilized an ordinal classifier, ordering lesions based on aggressiveness as within Gleason grading [62]. Brunese et al., Chaddad et al., and Jensen et al. utilized a ML pipeline for lesion grading, in which radiomic features are used for lesion categorization into different ISUP grades [63,64,65].

Only the study of Jensen et al. included multiple datasets from various sites [65]. Due to the smaller cohorts, two studies utilized cross-validation methods for validation. Both Chaddad et al. and Jensen et al. utilized independent data for algorithm validation [64,65]. No included study investigated additional value of CADx in a clinical setting and solely provided stand-alone performance of the algorithm (efficacy level 2).

### 4.2. Lesion Detection (CADe)

Besides classification of predefined ROIs on prostate MRI, several algorithms have been described that automatically detect suspicious PCa lesions (CADe). The general pipeline for these algorithms is displayed in Figure 6. Compared to lesion classification algorithms, no prior lesion annotation is necessary for classification as the AI method classifies the image on a voxel-level compared to a ROI. For this reason, PCa detection algorithms could aid in automated prostate MRI assessment, by presenting suspicious areas with probability maps and or segmentations to the reader. Studies described detection algorithms for two classes (e.g., malignant versus benign) and multi-class detection, in which malignant tissue is detected and classified according to its aggressiveness.

#### 4.2.1. Two-Class Lesion Detection

In total, seventeen studies for two-class lesion detection were included (Table 5).

Of these seventeen studies, six studies (35%) used conventional ML techniques, whereas the largest group utilized DL (11/17 studies (65%)). Cohort sizes ranged from 16 to 360 patients (median = 163). Ground truth was provided by either prostate biopsy (11/17 studies (65%)), radical prostatectomy (3/17 studies (18%)), or a combination of both (3/17 studies (18%)). Lesion detection was either determined on a cut-off at ISUP ≥ 1 or ISUP ≥ 2.

Most of the studies validated the performance of an original algorithm, whereas five studies performed a new validation study on existing CADe applications, assessing the robustness and generalizability of the algorithm.

McGarry et al. performed a follow-up study on a previously reported radiology-pathology mapping algorithm for high-grade PCa detection [81]. The follow-up study showed introduction of variability in model performance using different pathologists to annotate lesions [80]. Schelb and colleagues simulated clinical deployment of a prior developed DL algorithm for detection of csPCa [84,85]. In this study, a new cohort of 259 patients was included for validation of the algorithm. Schelb et al. concluded that similar performance compared to PI-RADS assessment by radiologists was observed, i.e.,: sensitivity of 0.84 for PI-RADS ≥ 4 versus 0.83 for DL and that regular quality assurance of the model should be desired to maintain its performance [84].

Multiple studies investigated the added diagnostic value of CADe (efficacy level 3). Zhu et al. performed a study in which integration of CADe with structured MR reports of prostate mpMRI was evaluated [89]. A DL algorithm was trained to create probability maps of csPCa which were visualized during reporting of prostate MRI. The AUC increased from 0.83 to 0.89 with CADe assistance during reading. Furthermore, with CADe, 23/89 lesions were correctly upgraded versus 6/89 lesions incorrectly downgraded [89].

Gaur et al. and Greer et al. utilized multi-center studies to evaluate the additional value of CADe for PCa [73,76]. Readers were first asked to assess mpMRI sequences without AI assistance. For the second session, readers were instructed to perform PCa detection with probability maps created by CADe, in combination with the full mpMRI sequences. In 2020, Mehralivand et al. performed a second multi-center study utilizing the AI technique previously utilized by Gaur et al. [73,82]. Instead of probability maps, attention boxes for csPCa were provided to reduce the compromised interaction between the radiologists and the AI system. The lesion based AUC did not significantly increase with CADe assisted reading (0.749 for MRI and 0.775 for CADe assistance) [82].

#### 4.2.2. Multi-Class Lesion Detection

Three articles were included for multi-class detection (Table 6).

Study cohorts comprised 417, 162, and 48 patients, respectively. Ground truth was provided by either prostate biopsy (2/3 studies (67%)) or radical prostatectomy (1/3 studies (33%)).

Cao et al. detected and classified lesions in six grade groups: normal tissue, and ISUP 1 to ISUP 5 [92]. The authors implemented a multi-class DL algorithm with ordinal encoding incorporating both T2W and ADC images. Vente and colleagues described a 2D DL segmentation approach in which zonal masks were implemented along mpMRI [93]. Their work assigned different classes to lesions according to the probability of the output layer, with a higher ISUP group correlating to a higher probability. A quadratic-weighted kappa score of 0.13 was achieved, indicating the still difficult task for lesion detection combined with grading [93].

In the study of Winkel et al., an algorithm combining both detection and classification of lesions according to PI-RADSv2.1 was investigated [94]. In their work, a prototype DL-based CADe application was validated in a prospective PCa screening study involving 48 patients. The algorithm firstly detects lesion candidates, then reduces false positive candidates followed by a classification algorithm according to PI-RADSv2.1. Kappa statistics were applied to assess the AI solution agreement with PI-RADSv2.1 classification by radiologists. A weighted kappa of 0.42 was observed, showing moderate agreement with PI-RADS scoring [94].

All studies were assigned an efficacy level of 2. Although Winkel et al. utilized a prospective study design for validation of the AI performance, no combination of AI-assisted lesion detection within a clinical workflow was performed [94]. The prospective aspect of the study, providing unique validation data, does provide stronger evidence for the algorithm [95]. Both Cao et al. and Winkel et al. incorporated comparison with radiological assessment [92,94]. Cao et al. showed a non-significant difference between the radiologists (sensitivity of 83.9% and 80.7%) and their algorithm (sensitivity of 80.5% and 79.2%), for the detection of histopathology-proven PCa lesions and csPCa lesions [92]. Winkel et al. showed that both the AI technique and the radiologist were able to identify all biopsy-verified PCa lesions [94].

### 4.3. Commercial CAD Algorithms for Prostate MRI

For CAD to be used in clinical practice, it needs to be approved by local authorities. In the United States the Food and Drug Administration (FDA) clears medical devices and in Europe a CE mark is necessary. For prostate MR analysis there are now six products commercially available, of which three are FDA cleared, one is CE marked and two products are both [96,97] (Table 7). The aim of these products is to optimize the prostate reading workflow and/or enhance lesion detection. The most important AI-based feature in five of these products is prostate segmentation to acquire volumetric information and calculate PSAd (OnQ Prostate, Cortechs.ai; PROView, GE Medical Systems; Quantib Prostate, Quantib; qp-Prostate, Quibim). One product claims to provide an image level probability for the presence of cancer including heatmaps to aid the radiologist in PCa detection (JPC-01K, JLK Inc.). Only a single product describes AI based multi-class lesion detection (CADe), classifying lesion candidates according to PI-RADSv2.1 (Prostate MR, Siemens Healthineers). The performance of a prototype of this product was validated within the work of Winkel et al. (Table 6) [94]. Further scientific evidence on the performance or efficacy of the products is limited.

## 5. Discussion

In this review, we identified current AI algorithms for PCa lesion classification (CADx) and detection (CADe). The narrative showed that most of the recent work is performed on lesion classification using ML applications with radiomics features. The included studies showed large differences in cohort sizes, ranging from 18 to 499 patients (median = 162), with different approaches for validation of algorithm performance. Few studies show efficacy levels higher than level 2, illustrating the limited evidence of AI PCa applications on the clinical impact and utility.

The majority of the included studies (50/59 (85%)) describe the stand-alone performance of algorithms and evaluate diagnostic accuracy with the AUC (efficacy level 2). While the AUC provides the ability to benchmark different algorithms and reader performances, in order to assess improved treatment, physicians are more interested in AI performance benchmarked against experienced readers to assess clinical utility [95]. Only a few studies (9/59 (15%)) address an efficacy level 3 or higher by validating the CAD within a clinical workflow (efficacy level 3) or assessing the clinical utility of PCa CAD by implementation of a DCA (efficacy level 4). While Deniffel et al., Song et al., Takeuchi et al., Xu et al., and Zhang et al. did show slight net-benefit on reducing unnecessary biopsies, studies utilizing CAD in a clinical workflow did not show significant improvement in PCa characterization and or detection [46,47,53,55,56]. This is in concordance with a recent meta-analysis on ML classification of csPCa, where the authors did not find an improved PCa detection using CAD in a clinical workflow [98].

The reported performances from included studies require careful interpretation. For example, only a limited number of prospective studies were found and cohort sizes were relatively small, ranging from 18 to 499 study subjects (median = 162). It is questionable if these relatively small datasets are sufficient to train robust and generalizable AI. To illustrate, the prototype CADe system validated by Winkel et al. was trained on 2170 biparametric prostate MR examinations, obtained from eight different institutions. A commercially available AI-CADe system for breast cancer detection was trained on 189,000 mammograms, obtained from four different vendors [94,99]. In addition, regarding reproducibility and generalizability, a limited number of the included studies utilized external validation. Appendix A illustrate the observed heterogeneity in cohort sizes and validation approaches. From a clinical perspective, it is of utmost importance to validate the predictive performance of an algorithm on external data in an extensive cohort [95]. Although multiple studies utilized a split-sample approach, in which a subset of data was solely reserved for validation, and therefore could assess the validity of the algorithm, split-set validation does not provide an accurate assessment of the generalizability. Utilization of test data from different institutions and potentially different MR systems could provide more generalizable results [49,100]. Liu et al. systematically reviewed 82 articles regarding the diagnostic accuracy of DL algorithms for disease classification in medical imaging evaluated against health-care professionals [101]. In their work, a major finding was the limited amount of publications presenting externally validations and performance comparison with health-care professionals on the same samples [101]. Castillo et al. systematically reviewed literature regarding ML classification of csPCa on MRI. Their work confirms the lack of homogeneous reporting and external validation. The authors therefore advocate for prospective study designs to assess added clinical value on external and new patient data combined with standardized reporting methods [98]. A possible solution to introduce more comparable results is to set up a challenge in which the algorithms are all validated on the same dataset. In this way, validation and benchmarking of algorithms remains centralized. An example of this approach is the grand-challenge platform (https://grand-challenge.org, accessed on 3 May 2021), in which various challenges are introduced for AI-based medical imaging tasks.

Within this work, distinction between ML techniques and DL techniques was made. Especially in the classification group (CADx), a large subset of studies utilized ML combined with a radiomics workflow. Compared to DL, radiomics is often favored due to the transparency in features used to learn a classification task, as compared to the ‘black-box’ phenomenon observed within DL [18]. Although the radiomics pipeline facilitates transparency for AI decision making, and therefore could aid in trustworthy AI, robustness of these algorithms needs yet to be assessed in larger studies [102]. A limitation for the radiomics pipeline is the reproducibility of imaging data. Factors such as inter-reader agreement for manually selected ROIs, variability due to different vendors and imaging protocols, and high amounts of correlated and clustered features, may limit the reproducibility and generalizability of these models [102,103].

This review has several limitations. The first limitation is the inclusion period between 2018 and February 2021. Due to this criterion, a multitude of proof-of-concept studies regarding PCa classification utilizing radiomics features were included. However, studies published prior to 2018 on traditional ML techniques for distinguishing benign versus PCa and multi-class classification or detection according to tumor aggressiveness, were excluded. Our rationale was that multi-reader studies or extended validation studies on AI algorithms developed before 2018 would be observed in more recent years. This was also observed for multiple included studies, in which traditional ML algorithms developed prior to 2018 were evaluated in multi-reader studies between 2018 and 2021 [73,76,82,89]. Future work could include publications prior to 2018 to reduce bias towards radiomics algorithms and introduce more work on traditional ML algorithms. This, however, would also limit the relevance of the current scientific field on AI for PCa assessment. Secondly, due to the observed heterogeneity within the cohort sizes and validation approaches, no meta-analysis on performance assessment was implemented and no comparison between various methods, such as ML and DL could be addressed with adequate support. To overcome this, performance could be assessed by grouping various AI approaches weighting the corresponding cohort sizes and validation approaches. This, however, exceeded the aims and objectives of this review and can be addressed in future work.

Many of the AI features, such as PCa detection and diagnosis, studied in literature have not yet found their way into commercial products. Current commercial CAD applications are mostly focused on prostate segmentation and volumetrics to improve the workflow for reporting and only a single application was observed with AI supported detection and lesion classification. This finding provides insights into the gap between academic results and clinical practice and may partly explain the lack of evidence on the impact of AI in clinical practice. Recent studies on CAD prototypes from, e.g., Winkel et al. indicate the future direction of commercial CAD applications, in which AI solutions for classification and detection of PCa lesions are gaining interest. The same authors have performed a new study on CAD implementation, which has been published after the inclusion period of studies within this review, underlining the rapid development within this field [104]. It is expected that future work on PCa CAD applications for lesion classification and detection will continue, with initiatives to centralize and combine data from multiple institutions to increase generalizability and robustness of PCa CAD arising [105].

## 6. Conclusions

Multiple AI algorithms for PCa classification and detection are being investigated in current literature. Although stand-alone performance of the algorithms shows to be promising for future implementation in clinical workflow, work on the generalizability and robustness still needs to be performed to assess the clinical benefit and utility of AI in this field. Future work should focus on increased cohort sizes, external validation, and benchmarking performance with expert readers to aid the development of reproducible and interpretable AI. In addition, studies incorporating CAD within a clinical workflow are necessary to demonstrate clinical utility and will guide the next steps for PCa CAD applications.

## Figures and Tables

**Figure 1 diagnostics-11-00959-f001:**
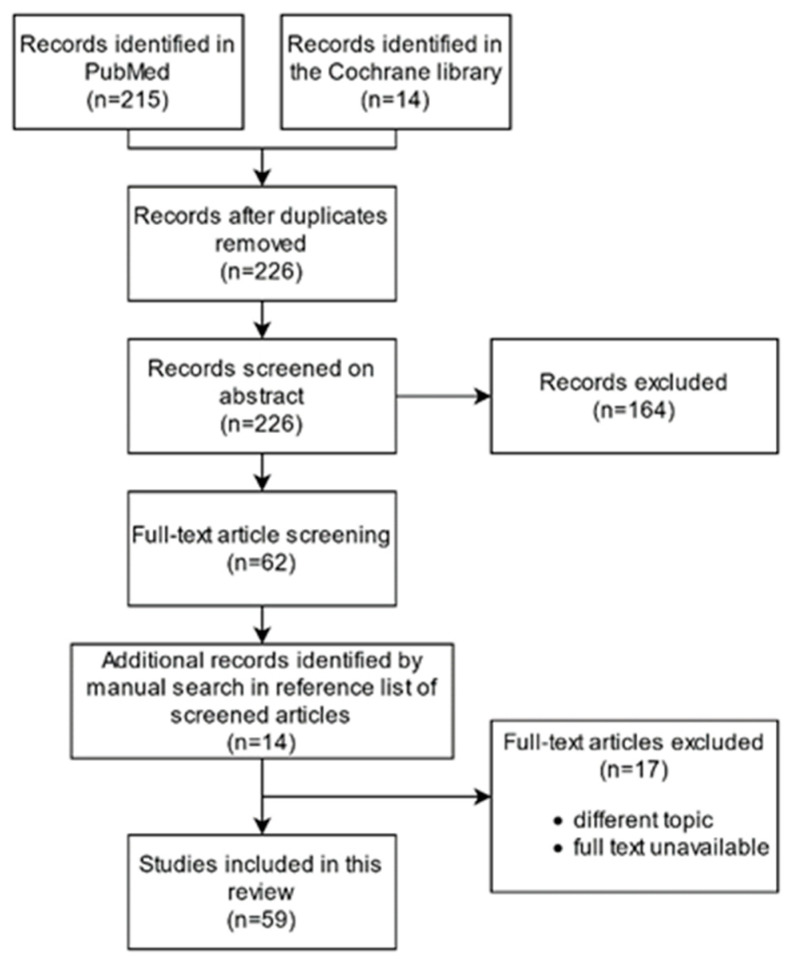
Flow diagram for search strategy.

**Figure 2 diagnostics-11-00959-f002:**
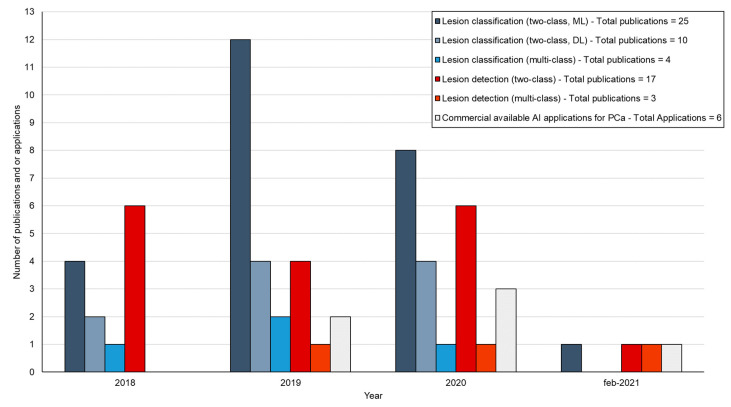
Overview of number of studies and commercially available AI applications for prostate MRI included in this review between 2018 and February 2021. Studies are categorized according to two-class lesion classification with machine learning (ML) and deep learning (DL), multi-class lesion classification, two-class lesion detection, and multi-class lesion detection. Most studies were observed for ML based two-class lesion characterization.

**Figure 3 diagnostics-11-00959-f003:**
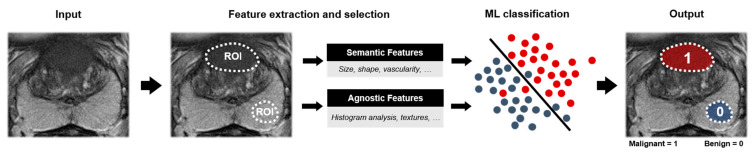
Machine learning (ML) workflow of two-class lesion classification for prostate cancer using an axial T2-weighted sequence. As input, multiparametric or singular MR sequences are used. Regions of interests (ROIs) are annotated, labeled, and used for feature extraction. A selection of features is used to train the ML-algorithm. As output, the annotated region is classified in one of the two classes.

**Figure 4 diagnostics-11-00959-f004:**
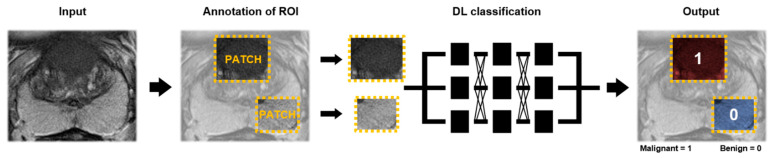
Deep learning (DL) workflow of two-class lesion classification for prostate cancer using an axial T2-weighted sequence. As input, multiparametric or singular MR sequences are used. On the MR images, regions of interest (ROIs) (patches or volumes) are annotated. The patches and/or ROIs are fed into the DL-algorithm. As output, a predicted label for the corresponding patch and/or ROI is provided.

**Figure 5 diagnostics-11-00959-f005:**
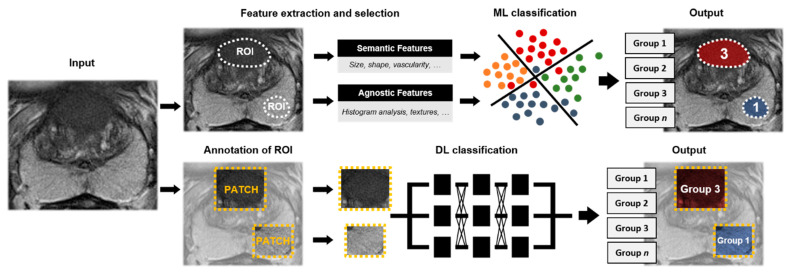
Workflow of machine learning (ML) and deep learning (DL) based multi-class lesion classification for prostate cancer using an axial T2-weighted sequence. The workflow follows a similar workflow as ML and DL pipelines described within two-class classification (see Figure 3 and Figure 4). As input, multiparametric or single MR sequences are utilized. Regions of interest (ROIs) are annotated and feature selection may be implemented prior to algorithm training. Classification is divided into multiple classes utilizing multiple labels within the ML and DL algorithm output. As output, annotations are graded according to the various labels (groups 1, 2, 3… *n*).

**Figure 6 diagnostics-11-00959-f006:**
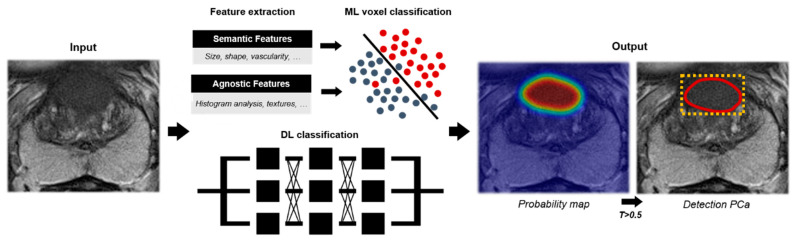
Deep learning (DL) and machine learning (ML) workflow of algorithms for two-class lesion detection for prostate cancer (PCa) using an axial T2-weighted sequence. As input, multiparametric or single MR sequences are utilized. During this, training features are trained and used to classify image voxels within benign or malignant classes. Algorithms provide a probability map for prostate cancer likelihood. Based on a threshold within the probability map (e.g., probability > 0.5), prostate cancer segmentations (red) or attention boxes based on prostate cancer segmentations (yellow) may be extracted.

**Table 1 diagnostics-11-00959-t001:** Hierarchical model of efficacy to assess the contribution of AI software to the diagnostic imaging process. An adapted model from van Leeuwen et al. [21], based on Fryback and Thornbury’s hierarchical model of efficacy [20].

Level	Explanation	Typical Measures
Level 1t *	Technical efficacy Article demonstrates the technical feasibility of the software.	Reproducibility, inter-software agreement, error rate.
Level 1c **	Potential clinical efficacy Article demonstrates the feasibility of the software to be clinically applied.	Correlation to alternative methods, potential predictive value, biomarker studies.
Level 2	Diagnostic accuracy efficacy Article demonstrates the stand-alone performance of the software.	Standalone sensitivity, specificity, area under the ROC ^¶^ curve, or Dice score.
Level 3	Diagnostic thinking efficacy Article demonstrates the added value to the diagnosis.	Radiologist performance with/without AI, change in radiological judgement.
Level 4	Therapeutic efficacy Article demonstrates the impact of the software on the patient management decisions.	Effect on treatment or follow-up examinations.
Level 5	Patient outcome efficacy Article demonstrates the impact of the software on patient outcomes.	Effect on quality of life, morbidity, or survival.
Level 6	Societal efficacy Article demonstrates the impact of the software on society by performing an economic analysis.	Effect on costs and quality adjusted life years, incremental costs per quality adjusted life year.

* Level 1t = Level 1, technical; ** Level 1c = Level 1, clinical; ^¶^ ROC = receiver operating characteristic.

**Table 2 diagnostics-11-00959-t002:** Overview of machine learning algorithms for two-class lesion classification of prostate cancer (PCa) between 2018 and February 2021. For classification categories of clinically significant (cs) and clinically insignificant (cis)PCa, ISUP grade is provided when available. Performance is indicated by the area under the ROC curve (AUC) when available, otherwise deviating performance metrics are included. Definition of efficacy levels is shown in Table 1.

Study	Input/Features	Algorithm	MR Sequences	Study Type (*n* = centers)	Cohort (Patients)	Validation Cohort/Total Cohort	Classification Categories	Ground Truth	AUC	Other Performance	Efficacy Level
Akamine, 2020 [22]	Quantitative MRI	HC	DWI, DCE	retrospective single center	52	N.A.	benign vs. PCa (not reported)	prostatectomy	-	Accuracy 96.3% (PZ) 97.8% (TZ)	2
Algohary, 2020 [23]	Intensity and texture features	QDA	T2W and ADC	retrospective multi center (4)	231	115/231	- low versus high risk PCa - low versus intermediate and high risk PCa (D’Amico Classification)	biopsy	0.87 (low vs. high risk PCa) 0.75 (low vs. intermediate-high risk PCa)	Accuracy (L vs. H) 53% (model) 48% (readers)	2
Antonelli, 2019 [24]	Quantitative MRI and intensity features	LR and NB	T2W, ADC and DCE	retrospective single center	164	30/164	cisPCa vs. csPCa (ISUP ≥ 2)	biopsy	0.83 (PZ) 0.75 (TZ)	Sensitivity at 50% threshold of specificity 88% (model) 82% (readers)	2
Bleker, 2020 [25]	Intensity and texture features	XGBoost	T2W, DWI, ADC and DCE	retrospective public dataset ^¶^	206	71/206	benign and/or cisPCa vs. csPCa (ISUP ≥ 2)	biopsy	0.870 [95%CI 0.980–0.754]		2
Bonekamp, 2018 [26]	Shape, intensity and texture features	RF	T2W, DWI and ADC	retrospective single center	316	133/316	benign and/or cisPCa vs. csPCa (ISUP ≥ 2)	biopsy	Lesion based 0.88	Sensitivity 97% (model) 88% (readers) Specificity 58% (model) 50% (readers)	2
Brancato, 2021 [27]	Shape, intensity and texture features	LR	T2W, ADC and DCE	retrospective single center	73	N.A.	benign versus PCa (ISUP ≥ 1)	biopsy	0.76 (PI-RADS = 3) 0.89 (upPI-RADS = 4) ^†^		2
Chen, 2019 [28]	Shape, intensity, and texture features	RF	T2W and ADC	retrospective single center	381	115/381	- benign versus PCa (ISUP ≥ 1) - cisPCa vs. csPCa (ISUP ≥ 2)	biopsy	ISUP ≥ 1 0.999 (model) 0.867 (readers) ISUP ≥ 2 0.931 (model) 0.763 (readers)		2
Dinh, 2018 [29,30]	Quantitative MRI and intensity features	Exponential model	ADC and DCE	retrospective single center	129	129 *	benign versus PCa (ISUP ≥ 2)	biopsy	0.95 [95% CI: 0.90–0.98] (CAD) 0.88 [95% CI: 0.68–0.96] (readers)		2
Ellmann, 2020 [31]	Quantitative MRI, shape, intensity, and clinical features	XGBoost	T2W, ADC and DCE	retrospective single center	124	24/124	benign vs. PCa (ISUP ≥ 1)	biopsy	0.913 (0.772–0.997)		2
Hectors, 2019 [32]	Intensity and texture features	LR	T2W, DWI and ADC	Retrospective, single center	64	N.A.	low vs. high risk PCa (ISUP ≥ 4)	prostatectomy	0.72		2
Kan, 2020 [33]	Quantitative MRI, shape, intensity, and clinical features	RF	T2W	retrospective multi center (2)	346	59/346 *	benign vs. PCa (ISUP ≥ 1)	biopsy	Lesion based 0.668		2
Kwon, 2018 [34]	Intensity and texture features	RF	T2W, DWI, ADC and DCE	retrospective public dataset ^¶^	344	140/344	benign and/or cisPCa vs. csPCa (ISUP ≥ 2)	biopsy	0.82		2
Li, 2018 [35]	Intensity features	SVM	IVIM, ADC, DCE	retrospective single center	48	N.A.	cisPCa vs. csPCa (ISUP ≥ 2)	biopsy	0.91 [95% CI: 0.85–0.95]		2
Liu, 2019 [36]	Intensity, texture, and filter features	LR	DCE	retrospective single center	40	N.A.	low vs. high risk PCa (ISUP ≥ 3)	biopsy	0.93		2
Min, 2019 [37]	Shape, intensity, texture, and filter features	LR (features) Linear model (radiomics signature)	T2W, DWI and ADC	Retrospective, single center	280	93/280	cisPCa vs. csPCa (ISUP ≥ 2)	biopsy	0.823 [95% CI: 0.67–0.98]		2
Orczyk, 2019 [38]	Quantitative MRI and intensity features	LR	T2W, ADC, and DCE	retrospective single center	20	N.A.	benign and/or cisPCa vs. csPCa (ISUP ≥ 2)	biopsy	0.93 [95% CI: 0.82–1.00]		2
Qi, 2020 [39]	Shape, intensity, texture, and filter features	RF and Multivariate LR (radiomics and clinical-radiological risk)	T2W, DWI and DCE	retrospective single center	199	66/199	benign vs. PCa (ISUP ≥ 1)	biopsy	0.902 [95% CI: 0.884–0.920] (model) 0.933 [95% CI: 0.918–0.948] (model with clinical-radiological variables)		2
Toivonen, 2019 [40]	Texture and filter features	LR	T2W, DWI and T2mapping	retrospective single center	62	N.A.	cisPCa vs. csPCa (ISUP ≥ 2)	prostatectomy	0.88 [95% CI: 0.82–0.95]		2
Transin, 2019 [29,41]	Quantitative MRI and intensity features	Exponential model	ADC and DCE	retrospective single center	74	74 *	benign and/or cisPCa vs. csPCa (ISUP ≥ 2)	biopsy and or prostatectomy	0.78 [95% CI: 0.69–0.87] (model) 0.74 [95% CI: 0.62–0.86] (readers)		2
Varghese, 2019 [42]	Texture features	Quadratic kernel based SVM	T2W and ADC	retrospective single center	68	N.A.	low versus high risk PCa (ISUP ≥ 4)	biopsy	0.71 [SE 0.01] (model) 0.73 (readers)		2
Viswanath, 2019 [43]	Intensity, texture, and filter features	QDA	T2W	retrospective multi center (3)	85	69/85 *	benign vs. PCa (not reported)	prostatectomy	Three sites validation 0.730, 0.686, 0.713		2
Woźnicki, 2020 [44]	Shape, intensity, texture, and clinical features	RF (benign vs malignant) SVM (csPCa vs cisPCa)	T2W and ADC	retrospective single center	191	40/191	benign vs. PCa (ISUP ≥ 1) cisPCa vs. csPCa (ISUP ≥ 2)	biopsy	ISUP ≥ 1 0.889 [95% CI: 0.751–0.990] (model) 0.779 [95% CI: 0.603–0.922] (readers) ISUP ≥ 2 0.844 [95% CI: 0.6–1.0] (model) 0.668 [95% CI: 0.431–0.889] (readers)		2
Wu, 2019 [45]	Shape, intensity, and texture features	LR	T2W and ADC	retrospective single center	90	N.A.	benign vs. PCa (ISUP ≥ 2)	prostatectomy	0.989 [95% CI: 0.9773–1.0000]		2
Xu, 2019 [46]	Intensity, texture, filter and clinical features	LR	T2W, DWI, and ADC	retrospective single center	331	99/331	benign vs. PCa (not reported)	prostatectomy	0.93 (model)		4 **
Zhang, 2020 [47]	Shape, intensity, and texture features	LR	T2W, DWI, and ADC	retrospective multi center (2)	159	83/159 *	cisPCa vs. csPCa (ISUP ≥ 2)	biopsy	0.84 [95% CI: 0.74–0.94]		4 **

HC = Hierarchical clustering. QDA = Quadratic discriminant analysis. LG = Logistic Regression. NB = Naïve Bayes. RF = Random Forest. SVM = Support Vector Machine. DWI = Diffusion weighted imaging. DCE = Dynamic contrast enhanced. ADC = Apparent diffusion coefficient. IVIM = Intravoxel incoherent motion. PZ = Peripheral zone. TZ =Transition zone. ^¶^ PROSTATEx database [48]. * Validation performed on an external dataset as compared to training. ** Efficacy level 4 was assigned for potential simulated therapeutic efficacy as determined with decision curve analysis. ^†^ upPI-RADS 4 = PI-RADS 3 lesions upgraded to PI-RADS 4 due to positive DCE-MRI.

**Table 3 diagnostics-11-00959-t003:** Overview of deep learning (DL) algorithms for two-class lesion classification of prostate cancer (PCa) between 2018 and February 2021. For classification categories of clinically significant (cs) and clinically insignificant (cis)PCa, ISUP grade is provided when available. Performance is indicated by the area under the ROC curve (AUC) when available, otherwise deviating performance metrics are included. Definition of efficacy levels is shown in Table 1.

Study	Input/Features	Algorithm	MR Sequences	Study Type (*n* = centers)	Cohort (Patients)	Validation Cohort/Total Cohort	Classification Categories	Ground Truth	AUC	Other Performance	Efficacy Level
Aldoj, 2020 [51]	MR: Spherical VOI lesion	CNN: 3D multi-channel	T2W, DWI, ADC and DCE	retrospective public dataset ^¶^	200	25/200	cisPCa vs. csPCa (ISUP ≥ 2)	biopsy	0.897 ± 0.008		2
Chen, 2019 [52]	MR: Patch lesion	Transfer Learning (CNN: Inception V3 and VGG-16)	T2W, ADC and DCE	retrospective public dataset ^¶^	346	142/346	benign vs. PCa (not reported)	biopsy	0.81 (InceptionV3) 0.83 (VGG-16)		2
Deniffel, 2020 [53]	MR: VOI prostate	CNN: 3D	T2W, DWI, and ADC	retrospective single center	499	50/499	benign and/or cisPCa vs. csPCa (ISUP ≥ 2)	biopsy	0.85 [95% CI: 0.76–0.97]	Sensitivity 100% (calibrated model) 84% (PI-RADS ≥ 4) 95% (PI-RADS = 3 + PSAd) Specificity 52% (calibrated model) 61% (PI-RADS ≥ 4) 35% (PI-RADS = 3 + PSAd)	4 **
Reda, 2018 [54]	MR: prostate segmentation and PSA	DL (SNCSAE)	DWI	retrospective single center	18	N.A.	benign vs. PCa (ISUP ≥ 1)	biopsy	0.98 [95% CI: 0.79–1]		2
Song, 2018 [55]	MR: Patch lesion	Deep CNN	T2W, DWI, and ADC	retrospective public dataset ^¶^	195	19/195	benign vs. PCa (not reported)	biopsy	0.944 [95% CI: 0.876–0.994]		4 **
Takeuchi, 2019 [56]	Intensity features and clinical variables	ANN: 5 hidden layers	T2W and DWI	retrospective single center	334	102/334	benign vs. PCa (ISUP ≥ 1)	biopsy	0.76		4 **
Wang, 2020 [57]	MR: Patch lesion	DL MISN (multi-input selec. Network)	T2W, DWI, ADC, and DCE	retrospective public dataset ^¶^	346	142/346	cisPCa vs. csPCa (ISUP ≥ 2)	biopsy	0.95		2
Yoo, 2019 [58]	MR: Patch prostate	Deep CNN with RF	DWI	retrospective single center	427	108/427	benign and/or cisPCa vs. csPCa (ISUP ≥ 2)	biopsy	Patient level 0.84 [95% CI: 0.76–0.91]		2
Yuan, 2019 [59]	MR: Patch lesion	Transfer learning (CNN: AlexNet)	T2W and ADC	retrospective single center and public dataset ^¶^	221	44 (20%)/221	cisPCa vs. csPCa (ISUP ≥ 2)	biopsy	0.896		2
Zhong, 2020 [60]	MR: Patch lesion	Transfer learning (CNN: ResNet)	T2W and ADC	retrospective single center	140	30/140	benign and/or cisPCa vs. csPCa (ISUP ≥ 2)	prostatectomy	0.726 [95% CI: 0.575, 0.876] (model) 0.711 [95% CI: 0.575–0.847] (readers)		2

CNN = Convolutional Neural Network. SNCSAE = Stacked nonnegatively constrained sparse autoencoder. ANN = Artificial Neural Network. RF = Random Forest. DWI = Diffusion weighted imaging. DCE = Dynamic contrast enhanced. ADC = Apparent diffusion coefficient. PSAd = Prostate specific antigen density. ^¶^ PROSTATEx database [48]. ** Efficacy level 4 was assigned for potential simulated therapeutic efficacy as determined with decision curve analysis.

**Table 4 diagnostics-11-00959-t004:** Overview of machine learning (ML) and deep learning (DL) algorithms for multi-class lesion classification of prostate cancer (PCa) between 2018 and February 2021. Performance is indicated by the area under the ROC curve (AUC) when available, otherwise deviating performance metrics are included. Definition of efficacy levels is shown in Table 1.

Study	Input/Features	Algorithm	MR Sequences	Study Type (*n* = centers)	Cohort (Patients)	Validation Cohort/Total Cohort	Ground Truth	AUC per Classification Category	Other Performance	Efficacy Level
Abraham, 2019 [62]	MR: Patch lesion	CNN: VGG-16. Ordinal Class Classifier	T2W, DWI and ADC	retrospective single public dataset ^¶^	112	N.A.	biopsy	ISUP 1 = 0.626 ISUP 2 = 0.535 ISUP 3 = 0.379 ISUP 4 = 0.761 ISUP 5 = 0.847	Quadratic weighted kappa 0.473 [95% CI: 0.27755–0.66785]	2
Brunese, 2020 [63]	Shape, intensity and texture features	Deep CNN	TW2	retrospective multiple public datasets ^¶¶, †^	72	N.A.	biopsy		Accuracy: normal = 0.96 ISUP 1 = 0.98 ISUP 2 = 0.96 ISUP 3 = 0.98 ISUP 4 = 0.97	2
Chaddad, 2018 [64]	Texture features	RF	T2W and ADC	retrospective single public dataset ^¶^	99	20 lesions / 40 lesions (per Gleason Group)	biopsy	ISUP 1 ≤ 0.784 ISUP 2 = 0.824 ISUP 3 ≥ 0.647		2
Jensen, 2019 [65]	Texture features	KNN	T2W, DWI, and ADC	retrospective single public dataset ^¶^	99	70 lesions / 182 lesions	biopsy	ISUP 1 = 0.87 (PZ), 0.85 (TZ) ISUP 2 = 0.88 (PZ), 0.89 (TZ) ISUP 1 + 2 = 0.96 (PZ), 0.83 (TZ) ISUP 3 = 0.98 (PZ), 0.94 (TZ) ISUP 4 + 5 = 0.91 (PZ), 0.87 (TZ)		2

CNN = Convolutional Neural Network. RF = Random Forest. KNN = k-nearest Neighbor. DWI = Diffusion weighted imaging. ADC = Apparent diffusion coefficient. PZ = Peripheral zone. TZ =Transition zone. ^¶^ PROSTATEx database [48]. ^¶¶^ PROSTATE-DIAGNOSIS database [66]. ^†^ Fused Radiology-Pathology Prostate Dataset [67].

**Table 5 diagnostics-11-00959-t005:** Overview of machine learning (ML) and deep learning (DL) algorithms for two-class lesion detection of prostate cancer (PCa) between 2018 and February 2021. Threshold for detection of PCa or clinically significant (cs)PCa is defined by the ISUP grade if applicable. Performance is indicated by the area under the ROC curve (AUC) when available, otherwise deviating performance metrics are included. Definition of efficacy levels is shown in Table 1.

Study	Input/Features	Algorithm	MR Sequences	Study Type (*n* = centers)	Cohort (Patients)	Validation Cohort/Total Cohort	Detection Threshold	Ground Truth	AUC	Other Performance	Efficacy Level
Alkadi, 2019 [70]	MR image	Deep CNN	T2W	retrospective single public dataset ^¶¶^	19 (2356 slices)	707 (30%)/2356 slices	PCa (not reported)	biopsy	0.995		2
Arif, 2020 [71]	MR image	Deep CNN	T2W, DWI and ADC	retrospective single center	292	194/292	csPCa (ISUP ≥ 2)	biopsy	0.65 (lesion > 0.03 cc) 0.73 (lesion > 0.1 cc) 0.89 (lesion > 0.5 cc)		2
Bagher-Ebadian, 2019 [72]	Texture and filter features	ANN: feed-forward multilayer perceptron	T2W, DWI and ADC	retrospective, single center	117	19/117 *	PCa (not reported)	biopsy	94%		2
Gaur, 2018 [73,74]	Shape, intensity, and texture features	RF	T2W, DWI and ADC	retrospective multi center (9) (5 centers data)	216	216 *	csPCa (ISUP ≥ 2)	biopsy and or prostatectomy	Patient level 0.831 (CADe) 0.819 (readers)		3
Gholizadeh, 2020 [75]	Intensity, texture, and filter features	SVM	T2W, DWI, ADC and DTI	retrospective single center	16	N.A.	PCa (ISUP ≥ 2)	biopsy	0.93 ± 0.03		2
Greer, 2018 [74,76]	Shape, intensity, and texture features	RF	T2W, DWI and ADC	retrospective multi center (8) (single center data)	163	163 *	csPCa (ISUP ≥ 2)	prostatectomy	PI-RADS ≥ 3 0.849 [95% CI: 79.0–89.5] (CADe) 0.882 [95% CI: 83.4–92.1] (readers)		3
Ishioka, 2018 [77]	MR image	CNN: Unet with ResNet50	T2W	retrospective single center	335	34/335	PCa (ISUP ≥ 1)	biopsy	Two validation 0.645, 0.636		2
Khalvati, 2018 [78]	Shape, intensity, and texture features	SVM	T2W, DWI, ADC, CDI	retrospective single center	30	N.A.	PCa (ISUP ≥ 1)	biopsy		Accuracy 86%	2
Lee, 2019 [79]	MR image	CNN: UconvGRU (2D image slices)	T2W, ADC and DCE	prospective single center (retrospective reading)	16	N.A.	csPCa (ISUP ≥ 2)	prostatectomy		F1 score: 0.5323	2
McGarry, 2020 [80,81]	Intensity features	Partial least-squares regression models	T2W, delta T1, DWI and ADC	retrospective single center	48	20/48	csPCa (ISUP ≥ 2)	prostatectomy	0.8 [95% CI: 0.66–0.90]		2
Mehralivand, 2020 [73,82]	Shape, intensity, and texture features	RF	T2W, DWI and ADC	retrospective multi center (5)	236	236 *	csPCa (ISUP ≥ 2)	biopsy and or prostatectomy	Lesion level 0.775 (CADe) 0.749 (readers)		3
Sanyal, 2020 [83]	MR image	CNN: U-net	T2W, DWI and ADC	retrospective single center	77	20/77	csPCa (ISUP ≥ 2)	biopsy	0.86 (ISUP ≥ 2) 0.88 (ISUP = 1)		2
Schelb, 2021 [84,85]	MR image	CNN: U-net	T2W, DWI and ADC	retrospective, single center	259	259 *	csPCa (ISUP ≥ 2)	biopsy		Sensitivity (PI-RADS ≥ 3, PI-RADS ≥ 4) 99%, 83% (model) 98%. 84% (readers) Specificity (PI-RADS ≥ 3, PI-RADS ≥ 4) 24%, 55% (model) 17%, 58% (readers)	2
Sumathipala, 2018 [86]	MR image	Deep CNN: Holistically Nested Edge Detector	T2W, DWI and ADC	retrospective multi center (6)	186	47/186	PCa (not reported)	biopsy and or prostatectomy	0.97 ± 0.01		2
Wang, 2018 [87]	MR image	CNN: dual-path multimodal	T2W, ADC	retrospective single center and public dataset ^¶^	360	N.A.	csPCa (ISUP ≥ 2)	biopsy	0.979 ± 0.009		2
Xu, 2019 [88]	MR image	CNN: ResNets	T2W, DWI and ADC	retrospective single public dataset ^¶^	346	103/346	csPCa (ISUP ≥ 2)	biopsy	0.97		2
Zhu, 2020 [89,90]	Intensity and texture features	ANN	T2W, DWI and ADC	retrospective, single center	153	153 *	csPCa (ISUP ≥ 2)	biopsy	0.89 [95% CI: 0.83–0.94] (CADe) 0.83 [95% CI: 0.76–0.88] (readers)		3

CNN = Convolutional Neural Network. ANN = Artificial Neural Network (ANN). RF = Random Forest. SVM = Support Vector Machine. DWI = Diffusion weighted imaging. DCE = Dynamic contrast enhanced. ADC = Apparent diffusion coefficient. DTI = Diffusion tensor imaging. CDI = Correlated diffusion imaging. CADe = Computer aided detection. * Validation performed on an external dataset as compared to training. ^¶^ PROSTATEx database [48]. ^¶¶^ I2CVB dataset [91].

**Table 6 diagnostics-11-00959-t006:** Overview of algorithms for multi-class detection of prostate cancer between 2018 and February 2021. Performance is indicated by the area under the ROC curve (AUC) when available. Definition of efficacy levels is shown in Table 1.

Study	Input/Features	Algorithm	MR Sequences	Study Type (*n* = centers)	Cohort (Patients)	Validation Cohort/Total Cohort	Ground Truth	AUC per Detection Category	Other Performance	Efficacy Level
Cao, 2019 [92]	MR images	CNN: FocalNet (multi-class)	T2W, ADC	retrospective single center	417	N.A.	prostatectomy	ISUP 2 ≥ 0.81 ± 0.01 ISUP 3 ≥ 0.79 ± 0.01 ISUP 4 ≥ 0.67 ± 0.04 ISUP 5 ≥ 0.57 ± 0.02		2
Vente, 2021 [93]	MR images and zonal masks	CNN: 2D U-Net	T2W, ADC	retrospective public dataset ^¶^	162	63/162	biopsy		Quadratic weighted kappa 0.13 ± 0.27	2
Winkel, 2020 [94]	MR images	Deep CNN: multi network	T2W, DWI and ADC	prospective single center (retrospective reading)	48	48 *	biopsy		weighted kappa (CADe with PI-RADS classification) 0.42 Lesion level Sensitivity PI-RADS 5 = 100% PI-RADS 4 = 73% PI-RADS 3 = 43%	2

CNN = Convolutional neural network. DWI = Diffusion weighted imaging. ADC = Apparent diffusion coefficient. CADe = Computer aided detection. * Validation cohort comprises an external dataset. ^¶^ PROSTATEx database [48].

**Table 7 diagnostics-11-00959-t007:** Commercially available AI applications for prostate MRI with FDA clearance and/or CE marking prior February 2021.

Company	Product	Key (AI) Features	Market Date	FDA	CE
Cortechs.ai	OnQ Prostate (previously RSI-MRI+)	prostate segmentation, enhanced DWI map	11–2019	510(k) cleared, Class II	
GE Medical Systems	PROView	prostate segmentation and volumetry, AI supported lesion segmentation, workflow optimization	11–2020	510(k) cleared, Class II	
JLK Inc.	JPC-01K	image level probability for cancer presence, heatmap/contour of malignancy location	04–2019		Class I
Quantib	Quantib Prostate	prostate segmentation and volumetry, AI supported lesion segmentation, workflow optimization	10–2020	510(k) cleared, Class II	Class IIb
Quibim	qp-Prostate	(regional) prostate segmentation and volumetry, workflow optimization	02–2021	510(k) cleared, Class II	
Siemens Healthineers	Prostate MR	prostate segmentation and volumetry, lesion detection and classification, workflow optimization	05–2020	510(k) cleared, Class II	Class IIa

## Data Availability

Not applicable.

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
