# Peer review of "Artificial Intelligence Based Algorithms for Prostate Cancer Classification and Detection on Magnetic Resonance Imaging: A Narrative Review"

_diagnostics, 2021, doi:10.3390/diagnostics11060959_

Round 1

Reviewer 1 Report

Dear Authors, 

the paper is well organized and written. 

The strenght and weak points of AI approaches are well documented. 

I appreciate the distinction between lesion detection and lesion classification.

I've nothing to add. 

Best Regards

Author Response

Dear reviewer,

We would like to thank you for your valuable time and feedback on our paper. Based on your input, we understand that no minor or major revisions are required.

Best regards,

Jasper Twilt

Reviewer 2 Report

Very well written article and adds valuable information to the reader. The authors cited articles that focus on identifying benign vs PCa but not on prognosis or grade of tumor. There might not be several papers out there, but there are a few that correlate the Gleason score with multi parametric MRI using machine learning models (eg PCA). While those articles are published before 2018, I think the authors should discuss this limitation in the discussion. 
Zhou et al. Incorporating Oxygen-Enhanced MRI into Multi-Parametric Assessment of Human Prostate Cancer. Diagnostics (Basel). 2017 Sep; 7(3): 48. doi: 10.3390/diagnostics7030048    

Author Response

Dear reviewer,

We would like to thank you for your valuable time and considerate feedback on our paper. We took your comment to mean that within the limitations, we did not clarify that we have excluded early work on multi-class tumor classification and or detection, correlating mpMRI features with histopathological grade.

To further clarify the limitation of missed articles concerning this topic we have edited and added the following lines in the discussion (lines 467-472):

"Due to this criterion, a multitude of proof-of-concept studies regarding PCa classification utilizing radiomics features were included. However, studies published prior to 2018 on traditional ML techniques for distinguishing benign versus PCa and multi-class classification or detection according to tumor aggressiveness, where excluded."

We look forward to hearing from you regarding our submission. We would be glad to respond to any further questions and comments that you may have.

Best regards,

Jasper Twilt

Reviewer 3 Report

Thank you for the opportunity to review this insightful paper.

The manuscript is a narrative review providing an overview of studies describing AI algorithms for prostate MRI analysis.

Differently and with a new approach compared to other reports, the authors focused AI algorithms for lesion classification and detection for prostate cancer, with a particular inclusion period (2018-2021).

The work represents a significant contribution to the field

The work is well organized and comprehensively described

The work is scientifically sound and not misleading

The references are appropriate, adequate and updated

The English used is correct and readable

The limitations are fully and extensively stated, including the lack and the need of external validation of the most of the studies. Moreover, the authors suggested potential perspectives in order to overcome the limitation they encountered, such as they enlighted the need to include investigations prior 2018 in order to reduce biases.

The conclusions are in line with the reported data and the discussion section.

Author Response

(The authors gave the same response as above.)
